# A New Cell Line Derived from the Spleen of the Japanese Flounder (*Paralichthys olivaceus*) and Its Application in Viral Study

**DOI:** 10.3390/biology11121697

**Published:** 2022-11-24

**Authors:** Yucong Yang, Yuqin Ren, Yitong Zhang, Guixing Wang, Zhongwei He, Yufeng Liu, Wei Cao, Yufen Wang, Songlin Chen, Yuanshuai Fu, Jilun Hou

**Affiliations:** 1Key Laboratory of Freshwater Aquatic Genetic Resources, Ministry of Agriculture, Shanghai Ocean University, Shanghai 201306, China; 2Shanghai Collaborative Innovation for Aquatic Animal Genetics and Breeding Genetics, Shanghai Ocean University, Shanghai 201306, China; 3Hebei Key Laboratory of the Bohai Sea Fish Germplasm Resources Conservation and Utilization, Beidaihe Central Experiment Station, Chinese Academy of Fishery Sciences, Qinhuangdao 066100, China; 4Yellow Sea Fisheries Research Institute, Chinese Academy of Fishery Sciences, Qingdao 266071, China

**Keywords:** *Paralichthys olivaceus*, cell line, virus susceptibility, Bohle virus, Lymphocystis disease virus, cell transfection

## Abstract

**Simple Summary:**

In recent years, with the continuous development of the mariculture industry, the scale of mariculture has gradually expanded, and the problem of mariculture diseases is prominent. Among them, viral diseases and bacterial diseases are the main concerns. Japanese flounder (*Paralichthys olivaceus*), as the main marine aquaculture fish, has high economic value, mainly distributed in China, Korea, and Japan. At present, the viruses that are extremely harmful to Japanese flounder are Lymphocytes disease virus (LCDV), Hirame rhabdovirus (HIRRV), Viral hemorrhagic septicemia virus (VHSV), et al., were difficult to control in the event of a large-scale epidemic, causing significant economic losses to the Japanese flounder culture industry. Fish cell lines are an important research tool with the advantages of low assay conditions, accurate control of assay results, and ease of operation, and thus have been widely used in virology research. We established a new cell line Japanese flounder spleen (JFSP) from the spleen tissue of the Japanese flounder (*P. olivaceus*). We explored the optimal growth conditions, chromosome morphology, and number and transfection experiment of JFSP cells *in vitro*. The susceptibility of JFSP cells to different viruses, including Bohle virus (BIV), Viral hemorrhagic septicemia virus (VHSV), Hirame rhabdovirus (HIRRV), Infectious hematopoietic necrosis virus (IHNV), and Lymphocystis disease virus (LCDV) and the expression changes of immune-related genes after virus infection were detected. The established JFSP cell line has enriched the fish cell resource bank and can be used as an excellent tool for genetic manipulation, studying host-virus interactions, and the development of potential vaccines.

**Abstract:**

A new cell line Japanese flounder spleen (JFSP) derived from the spleen of Japanese flounder (*Paralichthys olivaceus*) was established and characterized in this study. The JFSP cells grew rapidly at 29 °C, and the optimum fetal bovine serum concentration in the L-15 medium was 15%. Cells were subcultured for more than 80 passages. The JFSP cells have a diploid chromosome number of 2n = 68, which differs from the chromosome number of normal diploid Japanese flounder. The established cells were susceptible to Bohle virus (BIV), Viral hemorrhagic septicemia virus (VHSV), Hirame rhabdovirus (HIRRV), Infectious hematopoietic necrosis virus (IHNV), and Lymphocystis disease virus (LCDV), as evidenced by varying degrees of cytopathic effects (CPE). Replication of the virus in JFSP cells was confirmed by qRT-PCR and transmission electron microscopy. In addition, the expression of four immune-related genes, *TRAF3*, *IL-1β*, *TNF-α*, and *TLR2*, was differentially altered following viral infection. The results indicated that the cells underwent an antiviral immune response. JFSP cell line is an ideal tool *in vitro* for virology. The use of fish cell lines to study the immune genes and immune mechanism of fish and to clarify the immune mechanism of fish has important theoretical significance and practical application value for the fundamental prevention and treatment of fish diseases.

## 1. Introduction

Viral diseases cause massive death, resulting in huge economic losses for the aquaculture industry worldwide. Japanese flounder (*Paralichthys olivaceus*), as the main marine aquaculture fish, has high economic value, mainly distributed in China, Korea, and Japan. Diseases caused by viruses such as lymphocytes disease virus (LCDV) [1], hirame rhabdovirus (HIRRV) [2], viral hemorrhagic septicemia virus (VHSV) [3,4], chinook salmon bafinivirus (CSBV) [5] et al. were difficult to control in the event of a large-scale epidemic, causing significant economic losses to the Japanese flounder culture industry [6,7,8,9].

The fish cell line is widely used for fish virus isolation and research. Since the first fish cell line was established from the gonad of rainbow trout (*Oncorhynchus mykiss*) [10], there have been ~783 cell lines established from a variety of fish [11,12]. Previously, most fish cell lines have derived from freshwater and riverine anadromous fishes [13,14]. However, the expansion of research on marine fishes from the *in vitro* to the cellular level has led to the rapid development of marine fish cell lines in the last decade. In Japanese flounder, the cell lines, including the brain [15,16], fin [17], embryo [18], gill [19], skin [20], spleen [21], and kidney [22], have been established. However, with the increasing research demands, new cell lines from different tissues of the same species and from the same tissue of different species but have different genetic backgrounds are required to establish.

As an important hematopoietic and immune organ of fish, the spleen plays an irreplaceable role in specific and non-specific immunity of fish. This study aims to establish a new cell line, JFSP, derived from the spleen of Japanese flounder. Moreover, the study objective is to determine the optimal growth conditions, chromosome number, and transfection efficiency of JFSP. The JFSP cell susceptibility to different viruses from fish and amphibians will be assessed for the first time in this study. Additionally, the study will explore the immune-related gene expression patterns of JFSP cells upon viral infection.

## 2. Materials and Methods

### 2.1. Primary Cell Culture and Routine Maintenance

Healthy Japanese flounder (weight: 150 g, 30 cm in total length) was obtained from the Beidaihe Central Experiment Station, Chinese Academy of Fishery Sciences. The fish was narcotized with MS-222, exsanguinated with a syringe, and soaked with 75% alcohol. Spleen tissue was dissected in sterile Petri dishes and washed four times in Leibovitz’s L-15 (L-15, Gibco, Billings, MT, USA) medium containing 8% Penicillin-Streptomycin-Amphotericin B Solution (100×) (Solarbio, Beijing, China), and then immersed in Penicillin-Streptomycin-Amphotericin B Solution (100×) for 5 min. After washing, the minced tissues were dispersed into single cells using 0.25% trypsin solution for 30 min. The tissue fragments were inoculated into 25 cm^2^ cell culture flasks and cultured at 23 °C. A 5 mL complete culture medium containing L-15 medium, 15% fetal bovine serum (FBS, Gibco, Billings, MT, USA), 8% Penicillin-Streptomycin-Amphotericin B Solution (100×) (Solarbio, Beijing, China), 1% Japanese flounder serum, and Japanese flounder embryo extract, 2 ng/mL fibroblast growth factor (bFGF, Abcam, UK) and Leukemia Inhibitory Factor (LIF, Abcam, Cambridge, UK) was added to the culture flasks. One-half of the medium was changed every four days for three weeks. Observation of cell apposition and changes in cell morphology by inverted fluorescence microscope (DMi8, Leica, Wetzlar, Germany).

The old medium was removed, and the cell sheets were washed with Floating lotion containing L-15 medium, 8% Penicillin-Streptomycin-Amphotericin B Solution (100×) (Solarbio, Beijing, China) after the primary cell cultures had grown to confluence, and 90% of the culture flask bottom was covered with a monolayer of cells. The confluent cells were dispersed with 1 mL of 0.25% trypsin solution, subcultured at a ratio of 1:2, and maintained in an L-15 growth medium containing 15% FBS. The subcultured cells were passaged every 3–6 d. After the initial 10 subcultures, the growth medium was changed to L-15 with 10% FBS and reduced levels of antibiotics 1% Penicillin-Streptomycin-Amphotericin B Solution (100×) (Solarbio, Beijing, China).

### 2.2. Cryopreservation and Thawing of Cells

Japanese flounder (*P. olivaceus*) spleen (JFSP) cells were trypsinized, centrifuged, and then resuspended in an ice-cold storage medium (20% FBS, 10% dimethyl sulfoxide (DMSO, Sigma, St. Louis, MO, USA), and 70% L-15) at 2 × 10^6^ cells/mL for cryopreservation at every five passages. A 2 mL cell suspension was dispensed into 2 mL cryovials and then kept at 4 °C for 30 min, −20 °C for 1 h, and −80 °C overnight. They were then transferred to liquid nitrogen (−196 °C) for prolonged storage.

When the cells were resuscitated, the cryotubes were quickly removed from the liquid nitrogen tank, placed in a thermostat water bath at 37 °C, thawed by rapid shaking, and centrifuged at 1500 rpm for 5 min. Then the supernatant was removed and added with a fresh culture medium.

### 2.3. Species Confirmation of JFSP Cells

The partial sequence of the JFSP cell line’s 18s rRNA gene was analyzed to confirm its species. The genomic DNA of JFSP cells at the 70th passage was extracted using the TIANamp Genomic DNA Kit (TIANGEN, Beijing, China) according to the manufacturer’s instructions. The partial fragment sequence of the 18s rRNA gene was amplified using specific primers forward (5′-CCTGAGAAACGGCTACCACAT-3′) and reverse (5′-ATCCCGAGGTCCAACTACGAG-3′) (Accession Number: EF126037) designed according to the sequence available in the National Center for Biotechnology Information (NCBI) database. The 20 μL PCR reaction mixture contained 10 μL of the PrimeSTAR Max Premix (2×) (Takara, Kusatsu, Japan), 1 μL of each primer, 1 μL of the extracted genomic DNA, and 7 μL of the nuclease-free water. The optimum conditions for PCR include initial denaturation at 98 °C for 3 min, 35 cycles at 98 °C for 10 s, 60 °C for 5 s, and final elongation at 72 °C for 3 s. PCR products were analyzed on 1% agarose gel and sequenced using Sanger’s sequencing method.

### 2.4. Effects of Medium, Temperature, and FBS on Cell Growth

The effect of different FBS concentrations, temperatures, and culture medium on cell growth was examined in passage 55. JFSP cells were seeded at a density of 2 × 10^3^ cells per well in 96-well plates with different FBS concentrations (5, 10, 15, 20, and 25%), temperatures (11, 17, 23, and 29 °C), and culture medium (Self-made L-15, Business L-15, MEM, DMEM, DMEM/F12, and M199), then incubated for seven days to measure cell growth rate. Using the Cell Proliferation Kit (CCK-8, Biosharp, Hefei, China) and following the steps in the manufacturer’s instructions, the cell growth rate was estimated every 24 h after seeding.

### 2.5. Chromosome Number/Analysis

Chromosome analysis was performed using JFSP cells at subculture 65. The passage cells were cultured at 4 °C for 30 min before being transferred to a carbon dioxide cell incubator (3%, 23 °C). Colchicine with the final concentration of 0.5 or 1 μg/mL (Sigma, St. Louis, MO, USA) was added and incubated in the incubator for 5 h. Cells were digested with trypsin for 1 min and centrifuged at 1500× *g* rpm for 5 min. The cells were resuspended in 1 mL of 0.075 mol/L KCl solution at 37 °C for 40 min. Then it was fixed three times at 4 °C with Carnoy’s solution (the ratio of methanol to glacial acetic acid was 3:1) for 15 min. After centrifugation, the cells were suspended in 0.5 mL fixative solution, dripped onto −20 °C pre-cooled slides, and then immediately fire-roasted with the aim of dispersing the chromosomes. After staining with Giemsa solution (Sigma, St. Louis, MO, USA) according to the instructions, chromosomes were observed and counted under a light microscope.

### 2.6. Cell Transfection

Cells were inoculated into 12-well plates one day in advance until cell confluence reached 70 to 90% for subcellular localization experiments. According to Lipofectamine^TM^ 3000 Instruction manual (Invitrogen, Waltham, MA, USA), Opti-MEM (Gibco, Billings, MT, USA) diluted Lipo3000 reagent is used to mix well as reaction solution 1. The pEGFP-N1 premixture was prepared by diluting pEGFP-N1 with Opti-MEM, and the reaction liquid 2 was prepared by adding P3000. Then the reaction solution 1 and 2 were prepared into a plasmid-liposome complex according to the ratio of 1:1. After incubating for 15 min at room temperature, it was added to the cells. Finally, the mixture was added to cells, cultured at 23 °C for 2 d, and observed under a microscope.

### 2.7. Virus Susceptibility

JFSP cells susceptibility to Bohle virus (BIV), VHSV, HIRRV, infectious hematopoietic necrosis virus (IHNV), and LCDV were investigated. JFSP cells were seeded in 6-well plates and inoculated with 200 μL of virus inoculum for 1 h. The virus was removed and washed twice with L-15, then the infected cells were incubated with L-15 medium containing 3% FBS at 17 °C and subsequently observed for the cytopathic effect (CPE).

### 2.8. Preparation of Sample for Transmission Electron Microscopy (TEM)

JFSP cells at passage 65 were passaged in 75 cm^2^ cell culture flasks one day before. Until the cells reached 80% confluence, the cells were infected with 1 mL of virus solution and adsorbed for 1 h. The virus solution was removed, then washed twice and incubated at 17 °C for 3–4 d to observe CPE. The cells were collected when the CPE reached 70%; the medium was discarded, and the cells were fixed in room temperature electron microscope fixative (Servicebio, Wuhan, China) for 5 min and collected with a cell scraper (Biofil, Guangzhou, China), centrifuged at 2000× *g* rpm for 5 min. After discarding the supernatant, a new electron microscope fixative (Servicebio, Wuhan, China) was added, fixed at room temperature for 30 min, and stored at 4 °C. The cells were prepared for TEM analysis.

### 2.9. Detection of Viral Copies by qRT-PCR

JFSP cells were inoculated in 6-well plates and cultured at 23 °C. After one day of passaging, the cells reached about 90% fusion degree. The cells were washed with L-15 Floating lotion, inoculated with 200 μL of each virus, and then adsorbed for 1 h. The virus was removed, the cells were rewashed with 1 mL L-15 Floating lotion, and 2 mL of fresh maintenance medium containing 3% FBS was added. The cells were incubated at 17 °C for three days. Culture supernatants were collected from virus-infected cells at 1, 2, and 3 days post-infection (d p.i.) to estimate the replication efficiency of the virus. The viral DNA/RNA was extracted using the TIANamp Virus DNA/RNA Kit (TIANGEN, Beijing, China), the total RNA concentration was determined, and 1 ug was reverse transcribed into cDNA. The remaining sample was stored at −80 °C. 0.5 μL of the sample was used as a template for qRT-PCR experiments. Three replicates were set up, and the uninfected group was used as a negative control group. The virus plasmid DNA was diluted from 10^−1^ to 10^−7^ as a standard curve to count the number of copies according to the Cycle number Ct value. Viral loads of BIV, VHSV, HIRRV, IHNV, or LCDV were determined on a q225 qRT-PCR platform (Novogene, Beijing, China) using TB Green Premix Ex Taq II (Takara, Kusatsu, Japan) and primers (Table 1). The amplification conditions were 94 °C for 2 min, denaturation at 94 °C for 30 s, and extension at 60 °C for 20 s for 40 cycles. qRT-PCR was performed with three replicate wells each. After amplification, melting curve analysis ensured that the amplification products were single and that no dimers or secondary products were formed. The standard curve regression equation was obtained from the standard curve and Ct values, and the viral copy number was calculated and expressed as an average log10 copies/1 μg RNA.

### 2.10. qRT-PCR Analysis of Antiviral-Related Genes

Cell samples were collected from JFSP cells infected with the BIV, HIRRV, VHSV, IHNV, and LCDV for 36 h p.i., cellular RNA was extracted using the trizol method, and the concentration was measured and reverse transcribed into cDNA. The expression levels of the immune genes *TRAF3* and *IL-1β*, *TNF-α*, and *TLR2*, were examined in JFSP cells using TB Green Premium Ex Taq II (Takara, Kusatsu, Japan) on a q225 qRT-PCR platform (Novogen, Tianjin, China) to analyze the immune response to BIV, HIRRV, VHSV, IHNV, and LCDV infection. The primer pairs are listed in Table 1. The β-actin gene was used as the housekeeping gene. The cycle condition is initial denaturation 94 °C, 2 min. Then denatured at 94 °C for 30 s, annealed at 60 °C for 20 s, and cycled 40 times. Finally, it was extended at 65 °C for 5 s and finally extended at 95 °C for 5 s. The specificity of qRT-PCR products was analyzed according to the melting curve, and the relative expression level of immune genes was calculated by the 2^−ΔΔCt^ method [24].

## 3. Results

### 3.1. Primary Culture and Subculture

Primary cells migrating from spleen tissue fragments were observed on the fifth day and forming a monolayer within one month at 23 °C (Figure 1A). In the initial subcultures, the cells grow into radial fibroblasts, as well as epithelioid cells with flat, irregular polygons and round nuclei in the center. In the first 10 passages, the cells split at a ratio of 1:2 every 6–9 d and were cultured in the medium (containing 15% FBS and 8% antibiotics). With the increase of passage time, it was observed that fibroblasts decreased gradually, mainly manifested as epithelioid cells, and the cell morphology was relatively simple. After 10 passages, the cells were maintained in an L-15 medium (containing 10% FBS and 2% antibiotics) at a ratio of 1:2 every 3–6 d (Figure 1B,C). At present, the cell line has been passed on for over 80 passages and is named the JFSP cell line (Figure 1D).

### 3.2. Species Confirmation of JFSP Cells

The cell line was designated as the JFSP cell line after being subcultured for 80 passages. The partial fragment sequence of the 18s rRNA gene of the Japanese flounder was amplified, and the cDNA was extracted from cells as a template to authenticate the species of the JFSP cell line. The results showed that the PCR product length was 254 bp. Sequence analysis showed 100% sequence similarity of 18s rRNA between JFSP and *P. olivaceus*, indicating the originating of the established cell line (Figure 2A,B and Appendix A).

### 3.3. Effects of Temperature, FBS, and Medium, on Cell Growth

The results showed that JFSP cells could survive and grow at 11–29 °C in L-15 (Self-made) medium containing 15% FBS, with rapid growth at 29 °C (Figure 3A). JFSP cells could be maintained but not grown in 5% FBS at the same temperature, while the JFSP cell growth showed an increasing trend as the FBS content in the medium increased, with good growth in 15–30% FBS and cells reaching maximum growth on day 3 or 4 after culture. However, there was no significant difference between the cell growth amounts, 15% FBS was chosen as the optimum concentration for this experiment, considering the cost. (Figure 3B). M199 performed the best in terms of culture medium. L-15 (Self-made) and L-15 (Business) mediums maintained a stable growth rate. No significant differences existed in JFSP cell growth in MEM, DMEM, and DMEM/F12 (Figure 3C).

### 3.4. Chromosome Number

The chromosomal information of JFSP cells was analyzed at passage 65. The chromosome morphologies were mostly telocentric or submetacentric based on 137 metaphase plates (Figure 4A). The cell chromosome number ranged from 60–69, while most cells had 68 chromosomes (Figure 4B).

### 3.5. Cell Transfection

JFSP cell line was transfected with pEGFP-N1 using Lipofectamine^TM^ 3000 (Invitrogen) at passages 40 and 80. The expression of GFP in the JFSP cell line was observed by fluorescence microscope. It was found that the fluorescence expression could be detected at 12 h. Approximately 30% of the cells displayed green fluorescence after 24 h of transfection (Figure 5A–D). The results indicated that the JFSP cell line could express exogenous genes and be used for functional verification of genes *in vitro*.

### 3.6. Virus Susceptibility

JFSP cells were tested for susceptibility to different viruses. CPE was observed in JFSP cells within 48 h after BIV, VHSV, HIRRV, IHNV, and LCDV infection. Figure A is a control (Figure 6A). JFSP cells became rounded, and cell monolayers disintegrated after being inoculated with the BIV virus for 24 h. Cell refractive index increased after 48 h, revealing clusters of grape-like cells, cell shedding, and filamentous intercellular bridges (Figure 6B). VHSV infection resulted in vacuolated and granular cells, cell disintegration, and cell fragmentation after 48 h. Complete disintegration and rounding were observed after three days (Figure 6C). HIRRV showed a lesion effect characterized by the disintegration of cell monolayers and rounding of cells, followed by an increase in refractive index and the appearance of granular and irregular cells at 48 h of infection (Figure 6D). Cells exhibited irregular margins after 48 h of IHNV infection, followed by disintegration of cell monolayers, a small number of rounded cells, and partial abscission, with inconspicuous nuclei and filamentous interstitial bridges after three days of infection (Figure 6E). LCDV infection caused an initial lesion effect, followed by disintegration of cell monolayers and atrophy, with rounded cells and filamentous interstitial bridges after three days of infection (Figure 6F).

On completion of the qRT-PCR examination of virus replication in JFSP cells, the virus copy number in the cell supernatant increased with time. Figure 7A shows that the BIV copies in JFSP cells increased significantly from 3.72 to 6.84 log_10_ BIV RNA copies/μg at 3 days post-inoculation (d p.i.) (*p* < 0.01) (Figure 7A). Post-infected samples of VHSV, HIRRV, IHNV, and LCDV yielded similar results. The VHSV copies in JFSP cells increased from 4.64 to 6.82 log_10_ VHSV RNA copies/μg at 3 d p.i. (*p* < 0.01) (Figure 7B). The HIRRV copies significantly increased from 5.06 to 7.24 log_10_ HIRRV RNA copies/μg at 3 d p.i. (*p* < 0.01) (Figure 7C), while IHNV copies increased significantly from 4.40 to 5.54 log_10_ IHNV RNA copies/μg at 3 d p.i. (*p* < 0.05) (Figure 7D), and LCDV copies increased significantly from 3.34 to 3.80 log_10_ LCDV RNA copies/μg at 3 d p.i. (*p* < 0.05) (Figure 7F). The results showed that JFSP cells were susceptible to five viruses, and the virus could replicate through JFSP cells.

### 3.7. Virus Transmission Electron Microscopy Observation

JFSP cells inoculated with the BIV virus revealed an abundance of hexagonal, vesicular membrane virus particles with a diagonal diameter of 144 ± 8 nm (Figure 8A) in ultrathin electron microscopic sections. Mature virus particles are released outside the cell by budding through the cell membrane (Figure 8B). Numerous bullet-shaped ~165 × 70 nm viral particles proliferated from cell membranes and aggregated within cellular vesicles in VHSV-infected cells (Figure 8C,D). The virus replication and release in HIRRV-infected cells caused damage with severe vacuolization of the cytoplasm (Figure 8E). On the surface of cells, viral particles became aggregated. Meanwhile, some particles with vesicle encapsulation were discovered inside the cytoplasm (Figure 8F). The intact virion exhibited a bullet-shaped capsid enclosed with an envelope. Moreover, the virion has an average length of 160 nm and a width of 80 nm. The electron microscopy results of IHNV-infected cells revealed a large number of virus particles in the cytoplasm of 150–170 nm in length, 70–90 nm in width, and a typical elastic structure of the virus shape, as well as cells with severe vacuolization of the cytoplasm (Figure 8G,H). The JFSP cell line contained icosahedral LCDV particles. The diameter of the particles was approximately 160–300 nm (Figure 8I,J).

### 3.8. The Expression Profiles of Antiviral-Related Genes

qRT-PCR analyzed the immune genes, including *IL-1β*, *TRAF3*, *TNF-α*, and *TLR2* expression levels in JFSP cells after virus infection. *TRAF3* upregulation showed significant differences after 36 h of BIV infection (*p* < 0.001). *TRAF3* expression was significantly upregulated after HIRRV and VHSV infection (*p* < 0.05 or 0.01) but significantly downregulated after IHNV infection (*p* < 0.01) and downregulated after LCDV infection but not significantly different (*p* > 0.05) (Figure 9A). The *IL-1β* expression level was significantly upregulated to peak levels 36 h after VHSV infection (*p* < 0.05). BIV and HIRRV infections showed highly significant (*p* < 0.01), and IHNV infection showed extremely significant differences (*p* < 0.001), but infection with LCDV showed no significant difference from the control group (*p* > 0.05) (Figure 9B). *TNF-α* upregulation showed very significant differences after BIV and HIRRV-infected cells after 36 h (*p* < 0.01). The expression was significantly upregulated (*p* < 0.05) after VHSV infection. In contrast, *TNF-α* expression was extremely significantly downregulated after IHNV infection (*p* < 0.001) but downregulated after LCDV infection, not significantly (*p* > 0.05) (Figure 9C). The *TLR2* was significantly or extremely significantly downregulated after all viruses were infected 36 h (*p* < 0.01 or 0.001) (Figure 9D).

## 4. Discussion

In this study, we established and characterized a JFSP cell line from the spleen tissue of Japanese flounder (*Paralichthys olivaceus*). The optimum growth condition for JFSP cells was in L-15 medium with 15% FBS at 29 °C. Up to now, the JFSP cell line has been subcultured for more than 80 passages and cryopreserved at different passages. PCR analysis and sequence analysis showed 100% sequence similarity of 18s rRNA between JFSP and *P. olivaceus*.

Chromosomes are the carriers of genetic material, and each species has a specific chromosome number. Therefore, karyotype analysis is an important parameter for cell line identification [21]. The JFSP cell line has 68 chromosomes, which differs from the Japanese flounder (2n = 48). Other Japanese flounder cell lines, such as fin tissue from the Korean Sea flounder (KTS) cell line, had 46 chromosomes, which is less than the number of chromosomes in normal Japanese flounder [17]. Flounder fin (FFN) and flounder spleen (FSP) cell lines contain 64 and 62 chromosomes, respectively [21]. PoB1, PoB2, PoBf, and PoBh cell lines have chromosome numbers of 60–72 [16]. Other Japanese flounder cell lines, such as FF-11 [25], FG-9307 [19], FEC [13], JFSK_wt/FSK_alb [20], POBC [15], OFEC-17FEN [26], FGBC8 [27] and PoEKC [18] have a chromosome number of 48. Therefore, we speculate that there are differences in the number of chromosomes from different fish and different tissues of the same fish, not completely retaining the same number of chromosomes as the host species. One of the reasons for such phenomena is cell transformation. The cell transformation is the transformation of cells into an immortalized stage by genetic changes that directly involve changes in cellular DNA or genes, which can be inherited stably and passed on from generation to generation, resulting in changes in a range of biological properties and growth characteristics, are capable of long-term maintenance and reproduction. The cell line has been transformed from a restricted cell line to a continuous passaged cell line that can propagate and grow without restriction *in vitro*, resulting in immortalization. With the increase of culture time and the increase of passage times, cells may undergo a transformation that is somewhat related to their degree of evolution, possibly due to chromosome cleavage, superposition, or deletion resulting in doubling and deletion. It has been shown that during cell lineage transmission, the telomeres of chromosomes are progressively missing with the continuous transmission, which causes changes in chromosome number [21]. However, the detailed mechanism for chromosome number change during the *in vitro* culture of the cell need further study.

Cell line susceptibility and species specificity are essential for conducting studies on viral pathogen isolation, characterization, and propagation. However, it is clear that not every cell line can be used to identify all viruses. Different cell lines have different susceptibilities to viruses, so it is important to establish susceptible cell lines for different viruses. Several studies have been published on using Japanese flounder cell lines for virus isolation, including the gill cell FG, embryonic cell line HINAE, and kidney cell line for HIRRV isolation [2,22]. The Japanese flounder skin tissue cell line JFSK is sensitive to the viral neuro-necrosis virus (VNN) [20]. LCDV can be isolated using the Japanese flounder embryonic cell line FEC [13]. In contrast, Japanese flounder FFN and FSP cells showed cytopathic effects after inoculating IPNV, MABV, IHNV, spring viremia of carp virus (SVCV), and HRV [21]. VHSV, IHNV, HRRV, and Marine birnavirus (MABV) can infect the flounder fin tissue KTS cell line, but there is no CPE for LCDV and VNN [17]. However, new cell lines are required for virus propagation, viral vaccine preparation, and the study of virus-host cell interactions. In this study, the newly established cell line JFSP has susceptibility to LCDV, VHSV, HIRRV, and IHNV isolated from Japanese flounder but also to BIV. The BIV is an amphibian virus that has been found to infect a variety of frogs in artificial infection trials [28,29,30] and reptiles, as well as some fish, including Barramundi (*Lates calcarifer*) [31] and Tilapia (*Oreochromis mossambicus*) [32]. However, few susceptible cell lines have been used to isolate and identify BIV. Therefore, the JFSP cell line could be used as a tool in the study of BIV.

Viral infections predispose host cells to damage and necrosis, leading to innate immune response initiation by host pattern recognition receptors (PRRs) such as Toll-like receptors (TLRs) and retinoic acid-inducible gene I (RIG-I)-like receptors (RLRs) [23] and changes in a range of cytokines involved in the inflammatory response [33] including *TNF-α*, *IL-1β*, and *IL6*. Studies have revealed that fish infected with VHSV or IHNV have high *IL-1β* levels [34,35,36]. The present experiment verified at the cellular level that *IL-1β* is significantly upregulated after BIV, VHSV, HIRRV, and IHNV infections. TLR2 is a pattern recognition receptor [37,38] that induces inflammatory factors and interferon-stimulated genes to carry out antiviral immune responses. In this experiment, *TLR2* was downregulated after VHSV, BIV, HIRRV, HIRRV, and LCDV infections. TNF-α is a tumor necrosis factor, and TRAF3 acts as a tumor necrosis factor receptor that binds to TNF and participates in apoptosis and inflammatory responses together with antiviral functions. Studies showed that *TNF-α* expression was downregulated after LCDV infection in fish [39]. In this study, *TNF-α* and *TRAF3* were significantly downregulated after IHNV infection. LCDV infection showed a downregulation trend but not significantly, presumably due to the short duration of viral stimulation. In contrast, *TNF-α* was significantly upregulated after BIV, VHSV, and HIRRV infections. The study of the regulation of immune-related cytokine expression contributes to understanding cellular immunity levels and improves cell ability to resist resistance to pathogenic microorganisms.

## 5. Conclusions

In summary, we established a cell line JFSP from Japanese flounder spleen tissue that displayed susceptibility to fish viruses BIV, VHSV, HIRRV, IHNV, and LCDV. These characteristics make JFSP cells an excellent tool for studying exogenous gene expression, gene function validation, and host-virus interactions.

## Figures and Tables

**Figure 1 biology-11-01697-f001:**
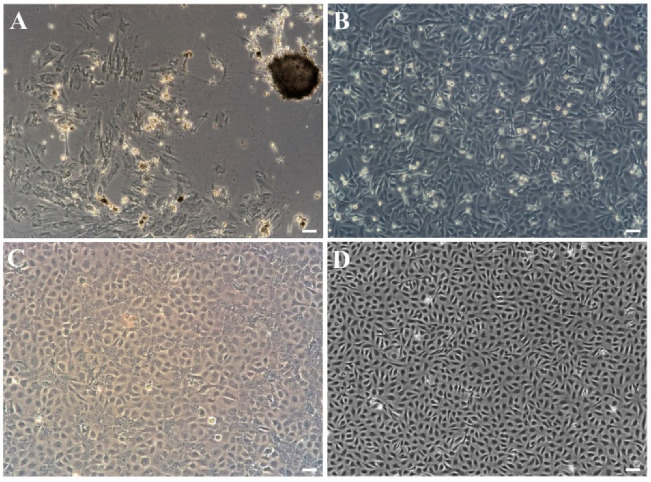
Morphology of the Japanese flounder spleen (JFSP) cells. (**A**–**D**) Morphology of JFSP cells at Primary passage (**A**), passage 40 (**B**), passage 60 (**C**), and passage 80 (**D**). Scale bar = 50 μm.

**Figure 2 biology-11-01697-f002:**
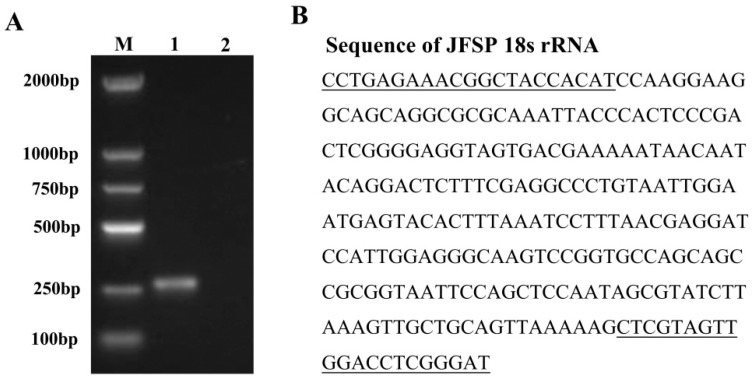
Agarose gel electrophoresis of PCR products from JFSP cells using specific primers for 18s rRNA. M, 2000 bp DNA maker; lane 1, 18s rRNA; lane 2, Blank.

**Figure 3 biology-11-01697-f003:**
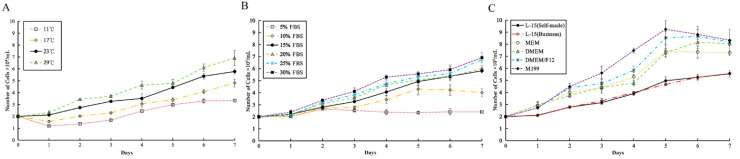
Growth kinetics of JFSP cell line at passage 55. (**A**) Effect of temperature on cell growth in L-15 medium containing 15% FBS. (**B**) Effect of L-15 medium with different concentrations of FBS on cell growth at 23 °C. (**C**) Effect of different media added with 15% FBS concentration on cell growth at 23 °C. Data are means ± SD of three measurements.

**Figure 4 biology-11-01697-f004:**
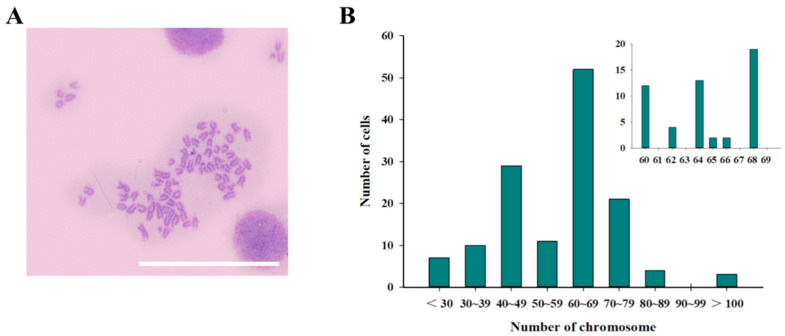
Chromosome morphology and frequency distribution of the number of entries in JFSP cells. (**A**) Chromosome morphology and (**B**) chromosome number statistics of JFSP cells treated with colchicine at 65 passage. Scale bar = 100 μm.

**Figure 5 biology-11-01697-f005:**
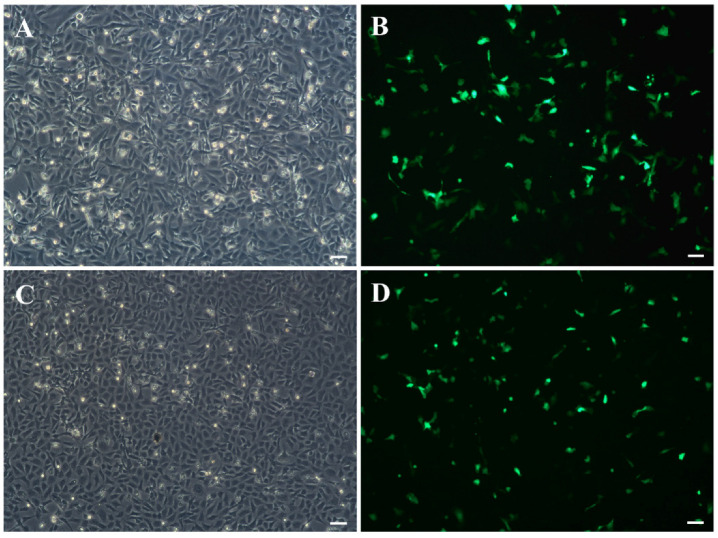
The expression of the GFP gene in JFSP cells 24 h after pEGFP-N1 transfection at passages 40 (**A**,**B**) and 80 (**C**,**D**). (**A**,**C**) Bright-field. (**B**,**D**) The expression of pEGFP-N1 in JFSP cells. Scale bar = 50 μm.

**Figure 6 biology-11-01697-f006:**
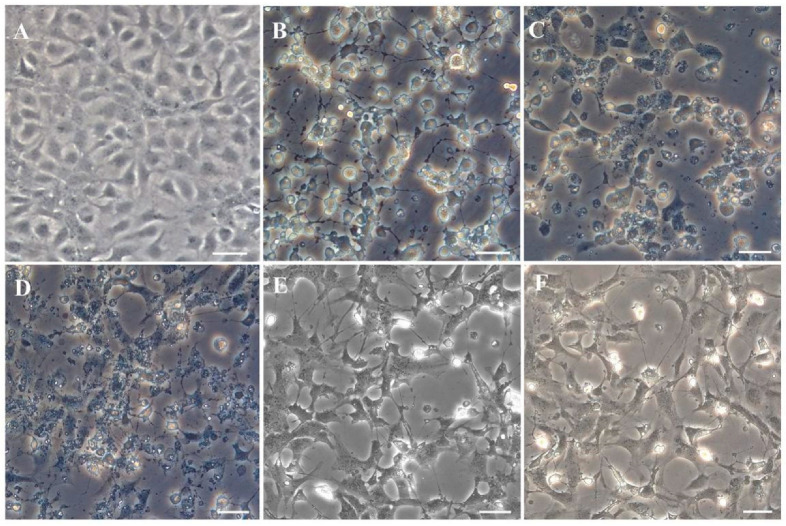
Observed CPE of JFSP cells infected by BIV, VHSV, HIRRV, IHNV, or LCDV through an inverted microscope. (**A**) Control group. CPE of JFSP cells infected with (**B**) BIV, (**C**) VHSV, and (**D**) HIRRV at 36 h p.i. (**E**) CPE of IHNV-infected JFSP cells at 48 h p.i. (**F**) CPE of LCDV-infected JFSP cells at 3 d p.i. Scale bar = 50 μm.

**Figure 7 biology-11-01697-f007:**
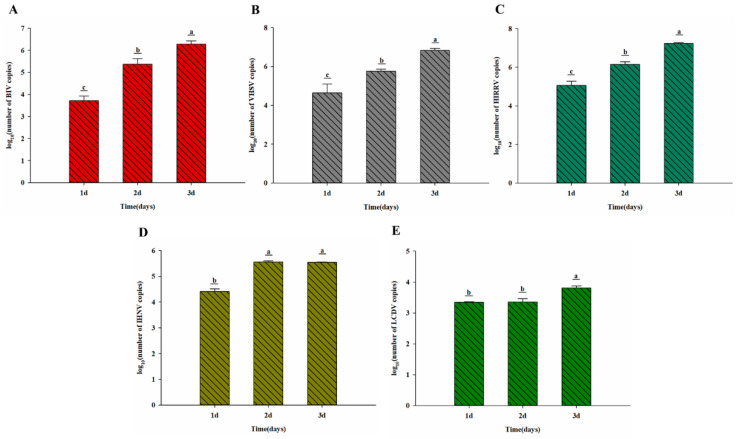
Replication characteristics of BIV, VHSV, HIRRV, IHNV, and LCDV in JFB cells. (**A**) BIV, (**B**) VHSV, (**C**) HIRRV, (**D**) IHNV, and (**E**) LCDV. Different letters denote significant differences among groups (*p* < 0.05).

**Figure 8 biology-11-01697-f008:**
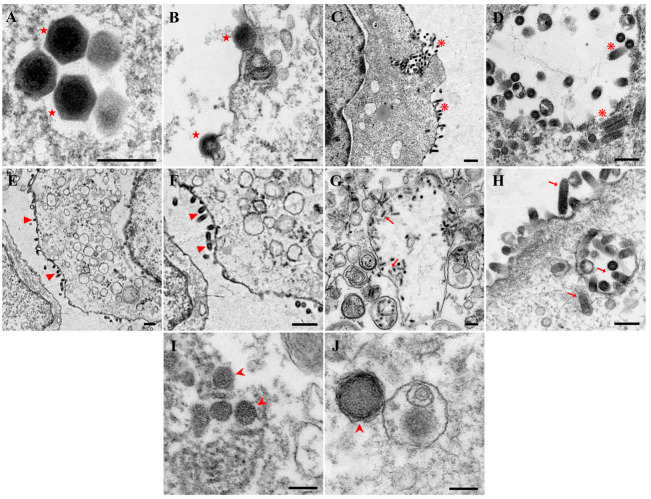
Ultrastructural observations of BIV, VHSV, HIRRV, IHNV, or LCDV-infected JFSP cells. Abundant virus particles were observed in BIV, VHSV, HIRRV, IHNV, or LCDV-infected cells. (**A**,**B**) Virus particles of BIV-infected cells. Scale bar = 200 μm. (**C**,**D**) Virus particles of VHSV-infected cells. (**C**) Scale bar = 500 μm. (**D**) Scale bar = 200 μm. (**E**,**F**) Virus particles of HIRRV-infected cells. Scale bar = 500 μm. (**G**,**H**) Virus particles of IHNV-infected cells. Scale bar = 200 μm. (**I**,**J**) Virus particles of LCDV-infected cells. Scale bar = 200 μm. ★: BIV, ❋: VHSV, ▶: HIRRV, ➞: IHNV, ➤: LCDV.

**Figure 9 biology-11-01697-f009:**
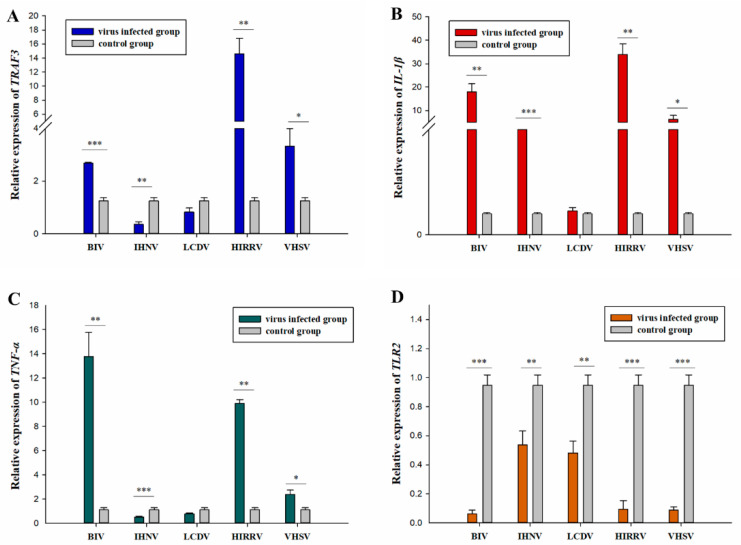
Expression of immune-related genes during infection of JFSP cells by BIV, VHSV, HIRRV, IHNV, or LCDV. (**A**–**D**) Relative expression levels of immune-related genes (TRAF3, IL-1β, TNF-α, and TLR2) in JFSP cells infected with the five viruses for 36 h p.i. Uninfected JFSP cells were used as a control group. All data are expressed as mean ± SEM (*n* = 3). * indicates a significant level of difference at *p* < 0.05, ** indicates a very significant level of difference at *p* < 0.01, and *** indicates an extremely significant level of difference at *p* < 0.001.

**Table 1 biology-11-01697-t001:** Primers used in this study.

Primer	Sequences (5′-3′)	Ta^a^	Source or Reference
**Genetic properties analysis**			
18s rRNA-F	CCTGAGAAACGGCTACCACAT	60	self-designed
18s rRNA-R	ATCCCGAGGTCCAACTACGAG		
**qRT-PCR analysis of viral replication**			
Q-LCDV-F	TTGACAGCAGGCGATTTAGAC	55	self-designed
Q-LCDV-R	AACGACGTTCCTCATTGGTTA		
Q-HIRRV-F	CGACCTGACATTCTACTATACGAC	55	self-designed
Q-HIRRV-R	TGGAGCACTTCCCTTCAATAA		
Q-IHNV-F	GACCCTTTGGGGATGAGTGG	60	self-designed
Q-IHNV-R	ATGCTCGTCTTGTACTGGGC		
Q-VHSV-F	GTCAAGGCAATTGTGGCTGG	60	self-designed
Q-VHSV-R	TGGAGTCAGTTTCCTCGTGC		
Q-BIV-F	AAAGTACATACTCTACCAGCTCCTC	55	self-designed
Q-BIV-R	GCTTCCAGTCTTTACGGTCAG		
**qRT-PCR analysis of immune-related genes**			
*TNF-α-F*	AAACACCTCACGTCCATCA	60	[23]
*TNF-α-R*	GCGTCCTCCTGACTCTTCT		
*IL-1β-F*	AAAGAAGCATCACCACTGTCT	60	[23]
*IL-1β-R*	TGGTAGCACCGGGCATTCT		
*TLR2-F*	GCTACATCTGCGACTCTCCT	58	self-designed
*TLR2-R*	CACAGGGACACGAACAAATC		
*TRAF3-F*	CACATCATTCCGCTCCTCTTA	60	self-designed
*TRAF3-R*	GCGTTCATTCACGACTTTACC		
*β-actin-F*	GGAAATCGTGCGTGACATTAAG	60	self-designed
*β-actin-R*	CCTCTGGACAACGGAACCTCT		

Ta^a^: annealing temperature (°C).

## Data Availability

All the data presented in this study are included in the article. If needed, Appendix A is available on request from the corresponding author.

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
