# Peer review of "A New Cell Line Derived from the Spleen of the Japanese Flounder (Paralichthys olivaceus) and Its Application in Viral Study"

_biology, 2022, doi:10.3390/biology11121697_

Round 1

Reviewer 1 Report

The fish cell line is widely used for fish virus isolation and research. Now there have been ~783 cell lines established from a variety of fish. In the present study, a new cell line JFSP derived from the spleen of Japanese flounder (Paralichthys olivaceus) was established and characterized. However, the manuscript should be much improved in logical organization before it is considered for publication.

1. There are many errors in grammar and syntax throughout the text of the manuscript, which are required for correction.

2. Line 24 and 33, the full name of JFSP should be given.

3. Line 25, Paralichthys olivaceus should be abbreviated.

4. Line 34, The JFSP cells grew rapidly at 23 ℃. 23℃ or 29℃?

5. Keywords should provide some topical contents of the paper.

6. Line 60-62, the cell lines from spleen have been established. Why establish a spleen cell line again? How does this cell line differ from previous cell lines?

7. Line 153-155, the sentence should be written.

8. Line 160, The JFSP cells grew rapidly at 29℃. The cells were infected with 1 mL of virus dilution solution incubated at 17℃ for 3–4 d  and observed for CPE. Why is 17℃ used in the virus infection experiment?

9. Figure 8. Viral particles are best marked with arrows.

10. Line 374-380, give the full names of KTS,FFN, FSP and so on?

11. The different cell lines of Paralichthys olivaceus have different number of chromosomes, and it needs to be verified.

12. There are duplicate references such as 17 and 36. Careful inspection is required.

Author Response

Response to Reviewer 1 Comments

Point 1: There are many errors in grammar and syntax throughout the text of the manuscript, which are required for correction.

Response 1: We apologize for the poor language for our manuscript. We worked on the manuscript for a long time and the repeated addition and removal of sentences and sections obviously led to poor readability. We have now worked on both language and resdability and have aslo involved native English speakers for language corrections. We really hope that the flow and language level have been substantially improved.

Point 2: Line 24 and 33, the full name of JFSP should be given.

Response 2: We have added the full name of JFSP (Japanese flounder spleen), and the detailed revision can be found in Line 26 and 35.

Point 3: Line 25, Paralichthys olivaceus should be abbreviated.

Response 3: We abbreviated Paralichthys olivaceus to P. olivaceus and the modification can be found in line 27.

Point 4: Line 34, The JFSP cells grew rapidly at 23 ËšC. 23 ËšC or 29 ËšC?

Response 4: We apologize for the impreciseness of the description. We have changed “23 ËšC” to “29 ËšC”, and the detailed revision can be found in Line 37.

Point 5: Keywords should provide some topical contents of the paper.

Response 5: We appreciate your suggestion to add cell line and cell transfection to the keywords related to the topic of the paper, which can be found in lines 50-51.

Point 6: Line 60-62, the cell lines from spleen have been established. Why establish a spleen cell line again? How does this cell line differ from previous cell lines?

Response 6: We describe the peculiarities that distinguish JFSP from other Japanese flounder spleen cells. As far as we known, we have for the first time assessed the immunological characteristics and transfection efficiency of Japanese flounder spleen cell line.

Point 7: Line 153-155, the sentence should be written.

Response 7: We made a simple modification that can be found in line 167-169.

Point 8: Line 160, The JFSP cells grew rapidly at 29 ËšC. The cells were infected with 1 mL of virus dilution solution incubated at 17 ËšC for 3–4 d  and observed for CPE. Why is 17 ℃ used in the virus infection experiment?

Response 8: The purpose of choosing 17 ËšC in our virus infection experiments is to enable the virus to better infest the cells. For example, LCDV has a high incidence at 10-20 ËšC and a significant self-healing phenomenon above 25 ËšC; while IHNV and VHSV viruses grow at 4-20 ËšC; HIRRV has an optimal growth temperature of 20 ËšC, and BIV can infect frogs and fish at 16-28 ËšC. Based on this system, we should also consider the appropriate temperature of cells. Therefore, a temperature of 17 °C was adopted as optimal.

Point 9: Figure 8. Viral particles are best marked with arrows.

Response 9: We have updated the Figure 8., and different tags describe different virus particles for infection.

Point 10: Line 374-380, give the full names of KTS, FFN, FSP and so on?

Response 10: Thank you for your suggestion, we made the appropriate changes to improve the rigor of the article, but not all names were given in full in the previous literature. Therefore, we believe that keeping the names uniform, abbreviations are also allowed.

Point 11: The different cell lines of Paralichthys olivaceus have different number of chromosomes, and it needs to be verified.

Response 11: Regarding this issue, it is true that subsequent verification of each of the other toothfish cell lines is needed to determine the finally conclusion. However, the point we are emphasizing in this experiment is not that all fish cell lines maintain the same chromosomes as their species, which we think is a reasonable fact. The presence of chromosome numbers that differ from the species chromosomes (2n=48) can be seen in line 376-386.

Point 12: There are duplicate references such as 17 and 36. Careful inspection is required.

Response 12: I sincerely thank you for reading it carefully, and I apologize for my lack of care. After careful examination, I will make changes in the article.

Reviewer 2 Report

This manuscript describes the establishment of a cell line from spleen of Japanese flounder. Although a cell line from the spleen of Japanese flounder already was reported in 2003 (not 2010 as listed in the reference list), the manuscript deserves to be published. However, there are some concerns that must be addressed.

Although English is not my native tongue, I think that the paper would benefit from improvement of the language. Also, there are some long paragraphs, with many words and not that much content (mostly in the discussion).

In the introduction, line 65, it is stated that the aim of the study was to establish a cell line form spleen of JF. But what is the long term goal? what problems can be solved by the use of this cell line? Vaccine development? Why focus on spleen? (And not on heart for example when a JF heart cell line is lacking?) The introduction should elucidate more extensively what challenges the JF farming is facing to justify / making the reader able to understand why the focus of the study. What types of diseases do the viruses you mention cause? what is their tropism?Do all target the spleen? the authors should say a bit more regarding the target organ of their research, the function, the anatomy and the different cell types.

In material and methods:

Line 10: In the heading change “origin” to species

Line 28: what is “self made L-15”?

Line 50: “cultured at 23”…for how long?

Line 52 and 69: “200ul virus”?? explain how the virus titration was done

Make sure that the terminology is used correctly (incubation, exposure, infection)

Line 58: not scanning electron microscopy? Transmission electron microscopy

Results:

Line 19: “Epithelioid cells” are activated macrophages resembling epithelial cells. Again, use correct terminology

Discussion:

Some of the paragraphs belong to the introduction as the authors are not discussing the findings but review what other groups have found (line 71-80). There are also many statements without references (example line 80-81). 

Author Response

Response to Reviewer 2 Comments

Point 1: In the introduction, line 65, it is stated that the aim of the study was to establish a cell line form spleen of JF. But what is the long term goal? what problems can be solved by the use of this cell line? Vaccine development? Why focus on spleen? (And not on heart for example when a JF heart cell line is lacking?) The introduction should elucidate more extensively what challenges the JF farming is facing to justify / making the reader able to understand why the focus of the study. What types of diseases do the viruses you mention cause? what is their tropism? Do all target the spleen? the authors should say a bit more regarding the target organ of their research, the function, the anatomy and the different cell types.

Response 1: The aquaculture of Paralichthys olivaceus is deeply affected by bacterial and viral diseases, so it is particularly important to study anti-disease vaccine. The use of fish cell lines to study the immune genes and immune mechanism of fish and to clarify the immune mechanism of fish has important theoretical significance and practical application value for the fundamental prevention and treatment of fish diseases. In addition, the transfection of foreign genes using fish cell lines has great application value in the fields of molecular biology and cell engineering, such as gene expression regulation, mutation analysis, protein purification and expression, and lays a foundation for the cultivation of new transgenic varieties with disease resistance and stress resistance. It can also study the mechanism of drug action, as well as vaccine development. As an important hematopoietic and immune organ of fish, spleen plays an irreplaceable role in specific and non-specific immunity of fish. LCDV can cause cysts in diseased fish. When the cyst cells rupture, a large number of virus particles pour into the tissue around the cyst tissue. As the blood circulatory system reaches the spleen, the phagocytes in the spleen are activated and begin to engulf the virus particles. When the immune system decreases, the spleen is infected, which in turn causes infection of other internal organs. IHNV can cause hemorrhage in fish and necrosis of hematopoietic tissue in kidney and spleen. HIRRV can cause bleeding of Japanese flounder, muscle tissues and organs, and necrosis of hematopoietic organs, which brings huge economic losses to fish culture. VHSV belongs to Mononegavirales, Rhabdoviridae and Novirhabdovirus. It can infect more than 80 species of fish, such as rainbow trout, flounder and turbot, causing hematopoietic organ necrosis and explosive death. It has brought huge economic losses to the aquaculture industry at home and abroad. BIV caused extensive necrosis and hyperemia of spleen tissue and complete destruction of interstitium.

Point 2: Line 10: In the heading change “origin” to species

Response 2: Thank you very much for your comments to make changes in the original text, which can be seen in line 117, 233, 236.

Point 3: Line 28: what is “self made L-15”?

Response 3: Self-made L-15 is we bought L-15 powder and then configured according to the instructions. We believe that there may be some differences between our own medium and the commercial medium. So in terms of cell culture, we wanted to see if there was a difference between the two, so we did a parallel experiment between the two.

Point 4: Line 50: “cultured at 23”…for how long?

Response 4: After transfection of pEGFP-N1 plasmid with lipo3000, fluorescence expression could be observed about 1-2 days later, but this kind of transfection was only transient, and the fluorescence would be gradually quenched if the transfection time was too long.

Point 5: Line 52 and 69: “200ul virus”?? explain how the virus titration was done

Response 5: We extracted from the culture supernatant of previously infected cells and did not determine the titer of the virus, because we are an identification experiment, not a quantitative experiment.

Point 6: Make sure that the terminology is used correctly (incubation, exposure, infection)

Response 6: Thank you very much for your suggestion. We have made changes to the relevant use in the article, see line 169, 173, 185, 285 for details.

Point 7: Line 58: not scanning electron microscopy? Transmission electron microscopy

Response 7: Thank you very much for your guidance on this issue. We have corrected it in the article.

Point 8: Line 19: “Epithelioid cells” are activated macrophages resembling epithelial cells. Again, use correct terminology

Response 8: Thank you very much for your suggestion. The main morphology of the cultured cells were epithelial cells, fibroblasts, migratory cells and polymorphic cells. The epithelial cells were flat, irregular and polygonal in shape, with a round nucleus in the center. Here I think that knowledge is similar in form, but not completely. The fibroblast type cell body is spindle or irregular triangle, with an oval nucleus in the center. I can make changes in the article, which can be found in lines 220-225.

Point 9: Some of the paragraphs belong to the introduction as the authors are not discussing the findings but review what other groups have found (line 71-80). There are also many statements without references (example line 80-81). 

Response 9: I'm sorry I didn't fully understand your point of view.We have searched the manuscript according to the mistakes you mentioned in the comments. “line 71-80” describes our operating procedures in the process of primary cell culture and subculture, so I don't think we can design the problems “the authors are not discussing the findings” you mentioned, and “Line 80-81” describes the ratio system of cell medium allocation in our laboratory. The matching system is similar but not exactly the same as that in similar articles (https://doi.org/10.1016/j.aquaculture.2022.738825) developed by our laboratory in the early stage, so I do not need to insert references. Although I did not fully understand your instructions, we have revised the whole text again, reviewed the references and changed the description of the discussion section as detailed in line 368-393. We hope our work can meet your requirements for the article. Thank you again for your review and help of this manuscript.

Reviewer 3 Report

In this article, a new cell line derived from the spleen of Japanese flounder (Paralichthys olivaceus) (JFSP) was established and characterized. The established cells were susceptible to Bohle virus, Viral hemorrhagic septicemia virus, Hirame rhabdovirus, Infectious hematopoietic necrosis virus, and Lymphocystis disease virus, viral copies and the immune-related gene expression in infected JFSP cells were detected. The established JFSP cell line provides an ideal tool for studying gene manipulation, host-virus interaction, and potential vaccine development. It is worthy of publication in Biology, but some issues should be resolved.

-- The abbreviation “JFSP” should be defined at the first appearance in Simple Summary,Abstract,and text. While the abbreviation FBS in Abstract is not necessary because it is only present once.

--In 2.4 “Effects of medium, temperature, and FBS on cell growth”, is there reference for temperature (11, 17, 23, and 29 ËšC)?

---Line 139: “Carnot's solution” should be “Carnoy's solution”

---Line 153-154: “viral hemorrhagic septicemia virus 154 (VHSV, HIRRV infectious hematopoietic necrosis virus (IHNV),” should be “VHSV, HIRRV, infectious hematopoietic necrosis virus (IHNV),”

--- why different culture temperature for JFSP was used in this study, for examples, line 151 “cultured at 23 Ëš C”; line 156: “the infected cells were kept at 17 ËšC”; line 160: “incubated at 17 ËšC”.

---Line 158: 2.8 Preparation of sample for scanning electron microscopy, however, in Line 166:” The cells were prepared for TEM analysis”, First, TEM should be defined; Second,”TEM” mean transmission electron microscopy, as shown in “3.7 Virus transmission electron microscopy observation” , but not scanning electron microscopy.

---In this study, a new cell line was established, as described by authors “over 90 passages and designated as the JFSP cell line” (Line 221), however, the genomic DNA of JFSP cells at the 70th passage was used for Origin confirmation of JFSP cells; The effect of different FBS concentrations, temperature and culture medium on cell growth was examined in passage 55; Chromosome analysis was performed using JFSP cells at subculture 65; JFSP cells at passage 65 were used for TEM. JFSP cell line was transfected with pEGFP-N1 at passages 40 and 80. Why different passages of JFSP cells were used in different experiment?

--- In Results, the author described “The cells have been subcultured for over 90 passages and designated as the JFSP cell line (Fig. 1D). but in Fig. 1D, the picture of passage 80 was shown.

---In Figure 2,lane 1 was not labeled; moreover, what is “lane B”?it should be explained.

---In Figure 3A, why didn't cells show logarithmic growth?

 Moreover, “(A) Effects of temperature on cell growth in L-15 that contained 15% FBS at 23 ËšC”, but temperature should be 11, 17, 23, and 29 ËšC.

---In Figure 6A was not cited in the text.

---In Discussion, Line 367-368:  the author described “The optimum growth condition for JFSP cells was in L-15 medium with 15% FBS at 23ËšC”, however, in Line 240 “with rapid growth at 29 ËšC”; in Line 246-247:”M199 performed the best in terms of culture medium”; line 247: “L-15 (Self-made) and L-15 (Business) mediums maintained a stable growth rate”. How to obtain the conclusion about the optimum growth condition?

Author Response

Response to Reviewer 3 Comments

Point 1: The abbreviation “JFSP” should be defined at the first appearance in Simple Summary,Abstract,and text. While the abbreviation FBS in Abstract is not necessary because it is only present once.

Response 1: Thank you for your careful guidance, I will make changes in line 26, 35 and 37.

Point 2: In 2.4 “Effects of medium, temperature, and FBS on cell growth”, is there reference for temperature (11, 17, 23, and 29 ËšC)?

Response 2: In line 135 it can be found that we have done experiments on the effect of different temperatures (11, 17, 23, and 29 ËšC) on cell growth.

Point 3: Line 139: “Carnot's solution” should be “Carnoy's solution”

Response 3: Thank you for the correction. I have corrected it in the article, and the detailed revision can be found in Line 149.

Point 4: Line 153-154: “viral hemorrhagic septicemia virus 154 (VHSV, HIRRV infectious hematopoietic necrosis virus (IHNV),” should be “VHSV, HIRRV, infectious hematopoietic necrosis virus (IHNV)”

Response 4: I sincerely apologize for my mistake and thank you for pointing out my mistakes and giving me suggestions for changes. You can find that I made corrections in line 165-166.

.

Point 5: Why different culture temperature for JFSP was used in this study, for examples, line 151 “cultured at 23 Ëš C”; line 156: “the infected cells were kept at 17 ËšC”; line 160: “incubated at 17 ËšC”.

Response 5: In pre-experiments performed for the current study, we have compared the suitability of JFSP cells for different temperatures. The results show that 23 ËšC is relatively suitable for cell growth, so in the subsequent cell passage process, we also set the temperature to 23 ËšC for cell growth. The infected cells are kept at 17 ËšC according to the susceptible temperature of the virus, at which several viruses can better infect the cells and have cytopathic effect.

Point 6: Line 158: 2.8 Preparation of sample for scanning electron microscopy, however, in Line 166:” The cells were prepared for TEM analysis”, First, TEM should be defined; Second,”TEM” mean transmission electron microscopy, as shown in “3.7 Virus transmission electron microscopy observation” , but not scanning electron microscopy.

Response 6: Thank you very much for your guidance. I have made changes in the article, which can be found in line 170 and have carefully studied the difference between transmission electron microscopy observation and scanning electron microscopy.

Point 7: In this study, a new cell line was established, as described by authors “over 90 passages and designated as the JFSP cell line” (Line 221), however, the genomic DNA of JFSP cells at the 70th passage was used for Origin confirmation of JFSP cells; The effect of different FBS concentrations, temperature and culture medium on cell growth was examined in passage 55; Chromosome analysis was performed using JFSP cells at subculture 65; JFSP cells at passage 65 were used for TEM. JFSP cell line was transfected with pEGFP-N1 at passages 40 and 80. Why different passages of JFSP cells were used in different experiment?

Response 7: JFSP cells were passed on according to the passage ratio at 1:2, and now they have been cultured for more than 80 passages. Different experimental time has led to different cell passages, but I have not been able to keep all the cells used in the experiment in the same algebra. The purpose of the transfection experiment in the 40 th passages and 80th passages is to determine whether there is a difference in the transfection efficiency of cells of different passages. In short, we believe that JFSP cell line maintained the biological characteristics of the spleen of the Japanese flounder regardless of the passage number.

Point 8: In Results, the author described “The cells have been subcultured for over 90 passages and designated as the JFSP cell line (Fig. 1D). but in Fig. 1D, the picture of passage 80 was shown.

Response 8: Thanks for your careful checks. We are sorry for our carelessness. In our resubmitted manuscript, the "passages" is revised, “80 passages” are correct.

Point 9: In Figure 2,lane 1 was not labeled; moreover, what is “lane B”?it should be explained.

Response 9: Thank you very much for your constructive comments. We have replaced the previous image and re-annotated it, which can be found in Line 243.

Point 10: In Figure 3A, why didn't cells show logarithmic growth?

Response 10: In Figure 3A, a part of data do not show an ideal logarithmic growth trend, but rather fluctuate. I think this kind of situation is very common, if very strict inspection, figure  3B and figure 3C presents the data is not entirely is the ideal logarithmic growth trend. This situation also often happens in published papers (e.g.: GAO et al., https://doi.org/10.1007/s11802-020-4435-z ). In our opinion, one of the main contributors to this discrepancies may be the change in instrument detection sensitivity for CCK8 assay as it is affected by the environment on the day of use. However, as a general trend, this is in line with expectation. Through this analysis, we successfully screened the optimum growth condition for JFSP cells was in L-15 medium with 15% FBS at 23ËšC.

Point 11: Moreover, “(A) Effects of temperature on cell growth in L-15 that contained 15% FBS at 23 ËšC”, but temperature should be 11, 17, 23, and 29 ËšC.

Response 11: Thank you very much for pointing out this detail, which I modified in line 257-258.

Point 12: In Figure 6A was not cited in the text.

Response 12: You can find that I supplemented your question at line 285-286, citing figure 6A.

Point 13: In Discussion, Line 367-368:  the author described “The optimum growth condition for JFSP cells was in L-15 medium with 15% FBS at 23ËšC”, however, in Line 240 “with rapid growth at 29 ËšC”; in Line 246-247: ”M199 performed the best in terms of culture medium”; line 247: “L-15 (Self-made) and L-15 (Business) mediums maintained a stable growth rate”. How to obtain the conclusion about the optimum growth condition?

Response 13: The optimum growth condition is 29 C, which I modified in line 371. At 0 to 5 days, the growth of JFSP cells in MEM showed no significant difference with that in M199, DMEM and DMEM/F12; however, from the 5th to 7th day, cellular growth slowed down, and JFSP cells showed a declined state. L-15 (Self-made) and L-15 (Business) medium could not reach the fastest growth rate but maintained steady growth.

Round 2

Reviewer 1 Report

The manuscript has reached publication requirements.

Author Response

Thank you very much for your affirmation of our outcomes.